# Comparison of the Effects of Chemokine Receptors CXCR2 and CXCR3 Pharmacological Modulation in Neuropathic Pain Model—*In Vivo* and *In Vitro* Study

**DOI:** 10.3390/ijms222011074

**Published:** 2021-10-14

**Authors:** Anna Piotrowska, Katarzyna Ciapała, Katarzyna Pawlik, Klaudia Kwiatkowski, Ewelina Rojewska, Joanna Mika

**Affiliations:** Department of Pain Pharmacology, Maj Institute of Pharmacology, Polish Academy of Sciences, 31-343 Krakow, Poland; anna.piotrowskamurzyn@gmail.com (A.P.); kat.ciapala@gmail.com (K.C.); pawlik@if-pan.krakow.pl (K.P.); kwiatkowski.klaudia@gmail.com (K.K.); rojewska@if-pan.krakow.pl (E.R.)

**Keywords:** neuroinflammation, chemokines, cytokines, neurodegeneration, glial cells

## Abstract

Recent findings have highlighted the roles of CXC chemokine family in the mechanisms of neuropathic pain. Our studies provide evidence that single/repeated intrathecal administration of CXCR2 (NVP-CXCR2-20) and CXCR3 ((±)-NBI-74330) antagonists explicitly attenuated mechanical/thermal hypersensitivity in rats after chronic constriction injury of the sciatic nerve. After repeated administration, both antagonists showed strong analgesic activity toward thermal hypersensitivity; however, (±)-NBI-74330 was more effective at reducing mechanical hypersensitivity. Interestingly, repeated intrathecal administration of both antagonists decreased the mRNA and/or protein levels of pronociceptive interleukins (i.e., IL-1beta, IL-6, IL-18) in the spinal cord, but only (±)-NBI-74330 decreased their levels in the dorsal root ganglia after nerve injury. Furthermore, only the CXCR3 antagonist influenced the spinal mRNA levels of antinociceptive factors (i.e., IL-1RA, IL-10). Additionally, antagonists effectively reduced the mRNA levels of pronociceptive chemokines; NVP-CXCR2-20 decreased the levels of CCL2, CCL6, CCL7, and CXCL4, while (±)-NBI-74330 reduced the levels of CCL3, CCL6, CXCL4, and CXCL9. Importantly, the results obtained from the primary microglial and astroglial cell cultures clearly suggest that both antagonists can directly affect the release of these ligands, mainly in microglia. Interestingly, NVP-CXCR2-20 induced analgesic effects after intraperitoneal administration. Our research revealed important roles for CXCR2 and CXCR3 in nociceptive transmission, especially in neuropathic pain.

## 1. Introduction

Despite numerous basic and clinical studies, the pathological mechanisms underlying neuropathic pain are still insufficiently understood, and, therefore, its treatment remains extremely difficult. Neuropathic pain is not completely relieved by conventionally used analgesics and therefore often becomes an ailment with no prospect of significant improvement [1,2]. Thus, studies focusing on recognizing new factors playing fundamental roles in the pathogenesis of neuropathic pain are important [3].

Cytokines, including chemokines, mediate and coordinate the neuroimmune interactions occurring after injury to the nervous system [4]. Recent evidence has highlighted an important role for chemokines in neuropathic pain development [4,5,6,7]. To date, only CC and CX3C subfamilies have been studied in detail under different neuropathic conditions, including studies from our group [7,8,9,10,11,12,13,14,15,16,17,18,19]. Chemokines from the CXC subfamily also appear to be critically involved in pain development and maintenance [5,20,21,22,23,24,25], but their exact roles remain to be evaluated. The CXC family is the second largest group of chemokines and comprises ligands of CXCR2 (CXCL1-3) and CXCR3 (CXCL4 and CXCL9-11), which are crucially involved in pathological nociceptive transmission, according to recent studies [5,20,21,22,24,26,27,28,29,30,31,32]. These proteins play a key role in the initiation and regulation of inflammation and are essential for neuronal transmission and cell-to-cell communication [33]. In the nervous system, CXCR2 is constitutively expressed on neurons and astrocytes [34,35], is upregulated in infiltrating macrophages and neutrophils [36,37] and in microglia activated in response to brain or nerve injury [34,38,39], and it is strongly associated with neutrophil chemotaxis [40,41,42]. CXCR3 is also expressed on various cell types, such as neurons, monocytes, lymphocytes (including CD4^+^ and CD8^+^ T and B cells), natural killer cells, and dendritic cells [20,42,43,44,45,46,47]. In recent years, extensive research has begun to identify roles of CXCR2 and CXCR3, other than the regulation of inflammation, in the context of nervous system pathologies, including neurological disorders [33,48,49,50]. The latest evidence showed a correlation between neuroinflammation involving CXCR2 and CXCR3 signaling and neuropathic pain [5,20,21,22,24]. Recent studies using various models of neuropathic pain strongly suggest that spinally located neuronal CXCR2 and CXCR3 are predominantly responsible for disturbed nociceptive transmission [21,22,24,28,51], although these receptors are also present on activated microglia and astrocytes; thus, these cells are known to be important for neuropathy development [21,22,34,44,52]. Importantly, we have already shown that intrathecal administration of CXCR2 (CXCL1-3) and CXCR3 (CXCL4, CXCL9-11) ligands quickly evokes strong pain-like behaviors in naive mice [21,22,53,54]. Interestingly, neutralization of endogenous chemokines, such as CXCL1-3 and CXCL9-10 [5,21,22,36,54], using specific neutralizing antibodies (nAbs) also exerts a beneficial analgesic effect on different animal models of neuropathic pain, such as spinal nerve ligation (SNL), partial sciatic nerve ligation (PSNL), and/or chronic constriction injury of the sciatic nerve (CCI). Based on these published data, CXCR2 and CXCR3 play an important role in neuropathic pain development. Therefore, we were prompted to verify the roles of CXCR2 and CXCR3 in the development of neuropathic pain and identify a new analgesic strategy based on blocking these receptors.

Our studies have been conducted to identify the similarities and differences between two chemokine receptors, CXCR2 and CXCR3, in neuroimmune processes occurring during neuropathy and to determine which one may become a better target for neuropathic pain treatment. In the present study, we used chronic constriction injury of the sciatic nerve as a neuropathic pain model to compare the analgesic properties and mechanism of action of CXCR2 (NVP CXCR2 20) and CXCR3 ((±)-NBI 74330)) antagonists. Assessments of thermal and mechanical hypersensitivity were performed after single and repeated intrathecal administrations of both substances in rats. Furthermore, we investigated the effects of intrathecal administration of both antagonists on changes in the mRNA and/or protein expression of immune factors important for the development and maintenance of neuropathic pain at the level of the spinal cord and DRG. In parallel, primary cultures of microglia and astroglia were used to test potential direct effects of the CXCR2 and CXCR3 antagonists on the nociceptive factors released by these cells. Finally, assessments of thermal and mechanical hypersensitivity were performed after a single intraperitoneal administration of CXCR2 and CXCR3 antagonists in mice to establish a better pharmacological target for creating effective drugs in the future.

## 2. Results

### 2.1. The Effects of Single Intrathecal Administration of (±)-NBI 74330 and NVP CXCR2 20 on Mechanical and Thermal Hypersensitivity on the 7th Day after Chronic Constriction Injury in Rats

Strong mechanical and thermal hypersensitivity were observed 7 days after chronic constriction injury in rats (Figure 1). A single *i.t.* administration of (±)-NBI 74330 (Figure 1A,B) or NVP CXCR2 20 (Figure 1C,D) dose-dependently diminished pain-related behaviors 1, 2, 4, and 6 h after treatment.

In the von Frey test, the strongest maximum possible effect was observed for (±)-NBI 74330 at a dose of 20 μg/5 μL (*p* = 0.0022) 4 h after administration and for NVP CXCR2 20 at a dose 20 μg/5 μL (*p* = 0.0002) 6 h after administration. In the cold plate test, (±)-NBI 74330 and NVP CXCR2 20 evoked the highest effects at a dose of 20 μg/5 μL 4 h after injection (*p* < 0.0001 for both). An analysis of the AUCs of the obtained data from the von Frey and cold plate tests showed that the most effective doses for both (±)-NBI 74330 and NVP CXCR2 20 at all-time points tested were 30 μg/5 μL (Figure 1E,F). Two-way ANOVA confirmed a significant interaction between the investigated CXCR3 and CXCR2 antagonists and the investigated time points in the cold plate test (*p* = 0.0015 and *p* < 0.0001, respectively).

### 2.2. The Effects of Repeated i.t. Administration of (±)-NBI 74330 and NVP CXCR2 20 on Mechanical and Thermal Hypersensitivity on the 7th Day after Chronic Constriction Injury in Rats

Chronic constriction injury evoked strong mechanical and thermal hypersensitivity, as measured using von Frey and cold plate tests (Figure 2A,B, respectively). The repeated administration of both (±)-NBI 74330 and NVP CXCR2 20 at a dose of 10 µg/5 µL reduced mechanical hypersensitivity (*p* < 0.0001 for both); however, a stronger effect was observed for (±)-NBI 74330 (*p* < 0.0001). In the cold plate test, the results were similar, as both CXCR3 and CXCR2 antagonists diminished thermal hypersensitivity (*p* < 0.0001 for both).

### 2.3. The Effect of Repeated Intrathecal Administration of (±)-NBI 74330 and NVP CXCR2 20 on Levels of the IL-1β, IL-18, and IL-6 mRNAs and Proteins in the Spinal Cord and DRG on the 7th Day after Chronic Constriction Injury in Rats

An analysis of mRNA expression in spinal cord showed that chronic constriction injury evoked a substantial upregulation of *IL-1β* (*p* < 0.0001) (Figure 3A), *IL-18* (*p* = 0.0227) (Figure 3B), and *IL-6* (*p* = 0.0014) (Figure 3C).

Moreover, both (±)-NBI 74330 and NVP CXCR2 20 were able to prevent CCI-induced changes in the levels of the *IL-1β* and *IL-6* mRNAs. No effects were observed on *IL-18* mRNA levels after substance administration. At the protein level, a significant upregulation of IL-1β (Figure 3D,G) was observed after CCI using Western blot (*p* = 0.0313) and Luminex (*p* < 0.0001) techniques. Additionally, 7 days after sciatic nerve injury, both (±)-NBI 74330 and NVP CXCR2 20 diminished the spinal levels of the interleukins measured using both methods. Interestingly, the CCI-induced increase in the level of IL-18 (Figure 3E,H) was decreased by CXCR3 and CXCR2 antagonists, as shown in the Luminex data (*p* = 0.0035). In addition, Western blot analysis showed that NVP CXCR2 20 may reduce IL-18 levels compared to vehicle-treated animals (*p* = 0.0667). Moreover, the increased level of IL-6 (Figure 3F,I) was significantly reduced by (±)-NBI 74330, as measured using Western blotting (*p* = 0.0703) and Luminex assays (*p* = 0.0824). No significant effect was observed after NVP treatment.

In the DRG, levels of the *IL-1β* (*p* = 0.0103) (Figure 3J), *IL-18* (*p* = 0.0064) (Figure 3K), and IL-6 mRNAs (*p* < 0.0001) (Figure 3L) were significantly upregulated after CCI. However, neither (±)-NBI 74330 nor NVP CXCR2 20 affected their levels. Western blot analysis of protein levels showed significant increases in IL-1β (Figure 3M), IL-18 (Figure 3N), and IL-6 (Figure 3O) levels after sciatic nerve injury. (±)-NBI 74330 was able to prevent the CCI-induced upregulation of IL-1β (*p* = 0.0845), IL-18 (*p* = 0.0002), and IL-6 (*p* = 0.0146), which was not observed in animals treated with NVP CXCR2 20.

### 2.4. The Effects of Repeated Intrathecal Administration of (±)-NBI 74330 and NVP CXCR2 20 on Levels of the IL-1RA, IL-18BP, and IL-10 mRNAs and Proteins in the Spinal Cord on the 7th Day after Chronic Constriction Injury in Rats

Chronic constriction injury did not alter levels of the *IL-1RA* (Figure 4A), *IL-18BP* (Figure 4B) or *IL-10* mRNAs (Figure 4C) in vehicle-treated animals; however, repeated (±)-NBI 74330 injections evoked an upregulation of the antinociceptive factors *IL-1RA* (*p* < 0.0001) and *IL-10* (*p* = 0.0204). Neither antagonist altered *IL-18BP* expression.

Significant differences in levels of the IL-1RA (Figure 4D), IL-18BP (Figure 4E) and IL-10 (Figure 4F) proteins measured using Western blotting were not observed between naive, vehicle-, (±)-NBI 74330- and NVP CXCR2 20-treated CCI-exposed animals.

### 2.5. The Effects of (±)-NBI 74330 and NVP CXCR2 20 Administration on Levels of the CCL2, CCL3, CCL4, CCL6, CCL7, CXCL4, CXCL9, and CXCL10 mRNAs in the Spinal Cord and DRG on the 7th Day after Chronic Constriction Injury in Rats

In the spinal cord, chronic constriction injury evoked a substantial upregulation of the *CCL2* (Figure 5A), *CCL3* (Figure 5B), *CCL4* (Figure 5C), *CCL6* (Figure 5D), *CCL7* (Figure 5E), *CXCL4* (Figure 5F), and *CXCL9* mRNAs (Figure 5G). After NVP CXCR2 20 administration, reduced levels of *CCL2* (*p* = 0.0009) and *CCL7* (*p* < 0.0001) were observed. Moreover, (±)-NBI 74330 injections reduced the levels of *CCL3* (*p* = 0.0389) and *CXCL9* (*p* = 0.0669) compared to vehicle-treated animals. Interestingly, both antagonists reduced the CCI-induced increases in the levels of *CCL6* (*p* = 0.0093) and *CXCL4* (*p* = 0.0184). No differences were observed for CXCL10 (Figure 5H).

In the DRG, CCI evoked an upregulation of the *CCL2* (*p* = 0.0301) (Figure 5I), *CCL4* (*p* = 0.0530) (Figure 5K), *CCL6* (*p* = 0.0169) (Figure 5L), and *CXCL9* mRNAs (*p* = 0.0986) (Figure 5O); however, the tested antagonists did not affect these changes compared to vehicle-treated rats. Additionally, increased levels of *CCL7* (*p* = 0.0660) (Figure 5M) were observed after the administration of CXCR2 and CXCR3 antagonists compared to naive cells. No significant changes were observed in the expression of *CCL3* (Figure 5J), *CXCL4* (Figure 5N) or CXCL10 (Figure 5P).

### 2.6. The Effects of (±)-NBI 74330 and NVP CXCR2 20 Administration on the mRNA and/or Protein Levels of IL-1β, IL-18, p-p38/p38 and pERK1/2/ERK1/2 in Primary Microglial Cell Cultures 1 or 24 h after LPS Stimulation

Levels of the *IL-1β* (Figure 6A, *p* < 0.0001) and *IL-18* mRNAs (Figure 6B, *p* = 0.0017) were substantially increased after LPS administration.

Both (±)-NBI 74330 and NVP CXCR2 20 downregulated IL-1β levels compared with the vehicle-treated LPS-stimulated group. Moreover, unstimulated (±)-NBI 74330- and NVP CXCR2 20-treated groups also showed reduced levels of this mRNA compared to vehicle unstimulated control. Similarly, both antagonists downregulated the level of the *IL-18* mRNA that was increased by LPS stimulation.

Substantial increases in the levels of IL-1β (Figure 6C), IL-18 (Figure 6D), and the p-p38/p38 (Figure 6E) and pERK1/2/ERK1/2 (Figure 6F) proteins were also observed in microglial cells from the vehicle-treated LPS-stimulated groups. (±)-NBI 74330 prevented the LPS-induced increase in the levels of IL-1β (*p* < 0.0001); however, no changes were observed when cells were pretreated with NVP CXCR2 20. Both antagonists diminished the level of IL-18, but a stronger effect was evoked by NVP CXCR2 20 (*p* = 0.0003). Moreover, CXCR3 and CXCR2 antagonists were also able to effectively reduce levels of p-p38/p38 protein (*p* < 0.0001), which were upregulated by LPS. In contrast, neither antagonist prevented the LPS-induced upregulation (*p* < 0.0001) of pERK1/2/ERK1/2 signaling.

### 2.7. The Effects of (±)-NBI 74330 and NVP CXCR2 20 Administration on Levels of the CCL2, CCL3, CCL4, CCL6, CCL7, CXCL4, CXCL9, and CXCL10 mRNAs in Primary Microglial and Astroglial Cell Cultures 24 h after LPS Stimulation

In the microglial cell cultures, (±)-NBI 74330 downregulated the expression of *CCL2* (*p* = 0.0117) (Figure 7A), *CCL3* (*p* = 0.0351) (Figure 7B), and *CCL4* (*p* = 0.0380) (Figure 7C). Additionally, both antagonists reduced *CCL6* (*p* = 0.0029) (Figure 7D), *CCL7* (*p* = 0.0019) (Figure 7E), *CXCL4* (*p* = 0.0019) (Figure 7F), *CXCL9* (*p* = 0.02289) (Figure 7G), and *CXCL10* (*p* = 0.00337) (Figure 7H) levels compared to vehicle-treated LPS-stimulated microglia.

In the astroglial cell cultures, decreased expression of *CCL4* (Figure 7C) (*p* = 0.0844) and *CXCL9* (*p* = 0.0079) (Figure 7G) was observed after (±)-NBI 74330 treatment compared to vehicle-treated LPS-stimulated astroglial cells. Neither (±)-NBI 74330 nor NVP CXCR2 20 influenced the levels of *CCL2* (Figure 7A), *CCL3* (Figure 7B), *CCL6* (Figure 7D), *CCL7* (Figure 7E), *CXCL4* (Figure 7F), or *CXCL10* (Figure 7H) in astrocytes. No differences were observed between groups vehicle-, NBI-, and NVP-treated of unstimulated microglial and astroglial cells (data not shown).

### 2.8. The Effects of a Single Intraperitoneal Administration of Different Doses of (±)-NBI 74330 and NVP CXCR2 20 on Mechanical and Thermal Hypersensitivity on the 7th Day after Chronic Constriction Injury in Mice

Strong mechanical and thermal hypersensitivity were observed 7 days after nerve injury in mice (Figure 8).

In the von Frey test, the single *i.p.* administration of NVP CXCR2 20 (Figure 8C,D) dose-dependently diminished pain-related behavior 1 and 3 h after treatment. The strongest maximum possible effect was observed for the dose of 10 mg/kg (*p* < 0.0001) 3 h after drug administration. In the cold plate test, NVP CXCR2 20 evoked the greatest effects at a dose of 10 mg/kg 1 h after injection (*p* < 0.0001). No analgesia was obtained after (±)-NBI 74330 treatment in either the von Frey or cold plate tests (Figure 8A,B). An analysis of the AUCs of the data obtained from the von Frey and cold plate tests showed that the most effective dose of NVP CXCR2 20 at all-time points tested was 10 mg/kg (Figure 8E,F). Two-way ANOVA confirmed a significant interaction (*p* < 0.0001) between the investigated CXCR2 antagonist and the investigated time points in the von Frey and cold plate tests.

## 3. Discussion

In the present study, we indicated that single and repeated intrathecal administration of CXCR2 (NVP CXCR2 20) and CXCR3 ((±)-NBI 74330) antagonists resulted in a strong and persistent dose-dependent analgesic effect 7 days after nerve injury in rats. Moreover, the repeated intrathecal administration of both antagonists regulated spinal mRNA and/or protein levels of pronociceptive cytokines, including interleukins (IL-1β, IL-18, and IL-6), on day 7 after injury, but only (±)-NBI 74330 also reduced the protein levels of the pronociceptive cytokines IL-1β, IL-18, and IL-6 in the DRG. Moreover, NVP CXCR2 20 effectively reduced the mRNA levels of *CCL2, CCL6, CCL7,* and *CXCL4*; in contrast, (±)-NBI 74330 diminished the levels of *CCL3, CCL6, CXCL4,* and *CXCL9,* which indicates slightly different mechanisms of action. Under neuropathic pain, we have previously shown that microglia and astroglia cells are strongly activated [8,17] and together with others we proved that both CXCR2 and CXCR3 are also present on these cells [21,22,34,44,52]. Our results obtained from primary microglial cell cultures suggest a direct effect of both antagonists on the release of pronociceptive cytokines by these cells (IL-1β, IL-18, CCL6-7, CXCL4, and CXCL9-10). In addition, (±)-NBI 74330 has a broader spectrum of activity, reducing the expression of CCL2-4. These results correspond well with previous studies showing the ability of (±)-NBI 74330 to modulate microglial activity in the spinal cord, explaining its superior effect on reducing mechanical hypersensitivity after multiple administrations for 7 days [22]. Importantly, despite the similar analgesic effects of both antagonists after intrathecal administration, substantial differences were observed after peripheral administration. The intraperitoneal injection of NVP CXCR2 20, but not (±)-NBI 74330, exerted antinociceptive effects on mice with fully developed symptoms of neuropathic pain.

Based on accumulating evidence, spinal levels of CXCR2 and CXCR3 increase as neuropathic pain develops [5,22,24]. Both receptors have been suggested to be similarly involved in neuroinflammatory and neurodegenerative diseases, including neuropathic pain [21,22,46,55]. In the present study, we showed that single intrathecal administration of the two highest doses of CXCR2 (NVP CXCR2 20) and CXCR3 ((±)-NBI 74330) antagonists to rats resulted in a strong, fast (observed already after one hour), and persistent analgesic effect for up to 6 h on day 7 after peripheral nerve injury, when neuropathic pain was fully developed. Their analgesic effects were very strong, approximately 80% MPE. The lowest doses of both antagonists induced no or little analgesia after a single administration; however, repeated injections were effective at reducing mechanical and thermal hypersensitivity measured on day 7 after sciatic nerve injury in rats. Notably, the CXCR3 antagonist (±)-NBI 74330 was more potent in ameliorating mechanical hypersensitivity, as measured by the von Frey test. Several mechanisms are involved in the development of mechanical hypersensitivity, including phenotypic changes in neurons and cells of the immune system [56,57]. Recently, microglial cells were considered essential in this mechanism. Activation of microglia is represented inter alia by increased expression of Iba-1 and by an increase in level of the phosphorylated form of p38 MAPK [58,59,60]. Downstream signaling pathways of p38 coordinate the production of inflammatory mediators, causing the sensitization of dorsal horn neurons of the spinal cord, which potentially leads to disturbances in excitability patterns that are related to neuropathic pain symptoms [56,58,61,62]. Moreover, the use of a p38 inhibitor prevents mechanical hypersensitivity caused by peripheral nerve ligation [61,63]. In our previous study, we showed that only an antagonist of CXCR3 ((±)-NBI 74330), but not CXCR2 (NVP CXCR2 20), reduced microglial activation [21,22]. Thus, these findings are the likely explanation for why the repeated intrathecal administration of (±)-NBI 74330 substantially reduced mechanical hypersensitivity after peripheral nerve damage [21]. The results of our research indicate a particularly important role for CXCR3 at the spinal cord level in the development of neuropathic pain.

Previous reports suggest that increased production of pronociceptive interleukins (e.g., IL-1β, IL-18, and IL-6) by immune and glial cells after peripheral nerve injury contributes to the development and maintenance of neuropathic pain [64,65,66,67]. Additionally, interleukins induce the expression of other pronociceptive mediators and are critically important in the regulation of immune responses [68]. Previous studies have shown an increase in spinal and DRG IL-1β levels after injury of the sciatic nerve [12,14], consistent with our results. In addition, in subjects with neuropathy, an increase in the spinal expression of IL-18 occurs that is parallel to the appearance of symptoms of neuropathic pain and the activation of microglial cells [60,69]. In the present study, we observed a decrease in IL-1β and IL-18 mRNA and/or protein levels in the spinal cord after (±)-NBI 74330 and NVP CXCR2 20 administration on day 7 after spinal nerve injury. Our previous results show that inhibition of their actions by the administration of an IL-1 receptor antagonist (IL-1RA) and IL-18 binding protein (IL-18BP) alleviates the symptoms of neuropathic pain and enhances the analgesic effect of opioids on neuropathy [60,70].

In our *in vitro* studies, we documented the direct effects of (±)-NBI 74330 and NVP CXCR2 20 on reducing IL-18 mRNA and protein levels in lipopolysaccharide-stimulated microglial cells. However, the reduction in spinal IL-1β expression induced by NVP CXCR2 20 does not seem to be related to direct microglial regulation, as we did not observe a decrease in the expression of this interleukin in LPS-stimulated microglial cells after treatment with the antagonist. This modulation may be related to neutrophils, since CXCR2 is involved in neutrophil infiltration into the spinal cord [71,72]. Neutrophils are the first inflammatory cells to reach the site of damage, peaking at 24 h after injury and declining after 3 days, although their numbers remain elevated [71,73]. In addition, neutrophils are known to produce pronociceptive mediators, including IL-1β [71,72], while microglia and macrophages promote the recruitment of neutrophils and participate in the cascade of events leading to neuropathy [74,75]. IL-6 is another important factor contributing to nociception; however, its action is not precisely defined, and available studies indicate a dual role. On the one hand, intrathecal administration of IL-6 in naive animals causes mechanical hypersensitivity [76], but in neuropathic pain states, it has analgesic properties [76,77]. Our research shows a spinal decrease in the levels of the IL-6 mRNA and protein after the administration of (±)-NBI 74330, consistent with our previous studies using the CCR5 antagonist maraviroc, which also effectively reduced pain and the activity of microglia and IL-6 release in parallel [12]. Interestingly, in the DRG, (±)-NBI 74330, but not NVP CXCR2 20, also reduced levels of the IL-1β, IL-18, and IL-6 proteins. These changes may be associated with the extinction of peripheral neutrophil activity on day 7 after nerve injury. In addition, macrophages and T lymphocytes also play an important role in the DRG, enhancing the response of satellite cells through the synthesis and release of cytokines, including IL-1β, IL-6, and CCL2, directly modulating sensory neurons and causing their ectopic discharges as a result of activation of receptors located on their surface [78,79]. CXCR3 is expressed at high levels on effector T cells; thus, its antagonist influences the immune factors that are produced by these cells [43]. Intracellular signaling pathways, such as p38 and ERK1/2, are involved in the regulation of the release of pronociceptive factors, and increases in their levels are observed in spinal microglia following nerve injury [62,80,81]. Other immunofluorescence studies have shown that the levels of the phosphorylated forms of p38 and ERK1/2 are particularly increased in microglial cells after nerve injury [80,82]. According to our previous paper, only (±)-NBI 74330 attenuates microglial activation and increases astrocyte activation [22]. Therefore, we decided to test the effects of both antagonists on primary glial cell cultures. However, our *in vitro* results indicate that both (±)-NBI 74330 and NVP CXCR2 20 influenced the level of p38 but not ERK1/2 in stimulated microglial cell cultures; therefore, each of these treatments influenced the release of pronociceptive factors to a different extent. We suggest that the observed differences might be because primary glial cell cultures do not completely reflect the complexity of neuroimmunological interactions at the levels of the spinal cord and higher pain pathways. Thus, the elucidation of the mechanisms underlying the multidirectional action of antagonists is an interesting goal of new research.

A different situation was observed for the modulation of antinociceptive factors in the spinal cord of rats with nerve injury after (±)-NBI 74330 and NVP CXCR2 20 administration. Protective functions in the pathogenesis of neuropathy have been attributed to IL-10. It is a conventional anti-inflammatory alternative polarity marker [12,83,84] and intrathecal administration of recombinant IL-10 reverses both mechanical and thermal hypersensitivity following nerve damage [85,86]. Additionally, IL-1RA and IL-18BP are important macrophage/microglia and astrocyte alternative polarization markers [12]. Recent studies suggest that the intrathecal administration of naturally occurring IL-1RA reduces the symptoms of neuropathic pain in rats after sciatic nerve injury [70]. An analgesic effect was also observed after administration of the IL-18 natural inhibitor IL-18BP [60]. Among the tested antagonists, only (±)-NBI 74330 increased the spinal expression of IL-1RA and IL-10 in rats that developed symptoms of neuropathic pain; however, it did not alter their protein levels on day 7 after nerve injury. Interestingly, our previous reports indicate that a CXCR3 antagonist reduces the levels of activated macrophage/microglial markers and increases the activation of astrocytes, which are believed to possess neuroprotective properties [87,88]. Thus, (±)-NBI 74330 appears to initiate a response that restores the balance of nociceptive factors and consequently may help maintain homeostasis during the development of the pathological process. Nevertheless, the effect of (±)-NBI 74330 on the expression of antinociceptive factors requires future in-depth research.

For further analysis, we chose chemokines (CCL2, CCL3, CCL4, CCL6, CCL7, CXCL4, CXCL9, and CXCL10) known to play a key role in nociception [8,13,18,22,53,89]. An intrathecal injection of most of these chemokines strongly induces mechanical and thermal hypersensitivity in naive animals [18,22,89,90]. Moreover, changes in their levels were noted in neuropathic pain states of different etiologies [8,13,19,22,91], consistent with our current results showing an increase in the levels of the CCL2, CCL3, CCL4, CCL6, CCL7, CXCL4, and CXCL9 mRNAs in the spinal cord and/or DRG on day 7 after sciatic nerve injury. Intrathecally administered antagonists of CXCR2 and CXCR3 were effective at reducing the levels of CCL2, CCL3, CCL6, CCL7, CXCL4, and/or CXCL9 in the spinal cord but not in the DRG. To date, one of the best-studied pronociceptive factors is CCL2. Most data indicate an increase in the spinal release of CCL2 by neurons and astrocytes after nerve injury [14]. Increased levels of CCL2 are also observed in the DRG, contributing to the activation of microglia via CCR2 in the spinal cord [8,92]. Furthermore, the intrathecal injection of CCL2 neutralizing antibodies not only reduces CCI-induced pain-related behavior in mice but also increases the analgesic effects of morphine and buprenorphine [89]. Importantly, in the present study, an intrathecally administered antagonist of CXCR2 diminished the *CCL2* mRNA level. CCL3 also causes hypersensitivity by directly sensitizing primary sensory neurons [93], and is responsible for disturbances in nociception processes caused by nerve injury and diabetes [94]. Moreover, neutralization of CCL3 reduces neuropathic pain symptoms [18]. In our study, (±)-NBI 74330 decreased the spinal level of *CCL3* mRNA. CCL6 and CCL7 are other chemokines that are important in the process of nociception, although relatively little is known about them. In 2020, we showed that CCL6 is important for the maintenance of neuropathic pain because it is upregulated beginning on day 7 after nerve injury and persists until 28 days [19]. Importantly, both antagonists substantially diminished the spinal level of this chemokine. CCL7 appears to be very important for both the initiation and maintenance of neuropathic pain, since its expression is substantially upregulated in the spinal cord and DRG shortly after injury and persists for up to 28 days [19]. Furthermore, CCL7 neutralizing antibodies not only reduce nerve injury-induced pain-related behavior but also increase the analgesic properties of morphine and buprenorphine [89]. Importantly, the intrathecally administered antagonist of CXCR2 diminished the spinal level of the *CCL7* mRNA. In a subsequent analysis, we also chose chemokines from the CXC group, which our previous study revealed to be involved in hypersensitivity occurring in the neuropathic pain model [21,22]. In 2018, we described CXCL4 upregulation in the spinal cord and DRG 7 days after nerve injury and its pronociceptive properties after intrathecal administration to naive mice [22]. Importantly, both antagonists, NVP CXCR2 20 and (±)-NBI 74330, diminished the spinal level of this chemokine. An intrathecal injection of CXCL9 also produces a strong pronociceptive effect [22]. Moreover, we have shown that CXCL9 is important for the maintenance of neuropathic pain because its upregulation in the spinal cord and DRG begins on day 7 and persists until 28 days after nerve injury. Additionally, CXCL9 neutralizing antibodies reduce the symptoms of neuropathic pain in mice [22]. Similarly, other authors described an increase in CXCL9 expression in the spinal cord and serum level of animal models of streptozotocin-induced diabetes [53,95], an autoimmune encephalomyelitis model [96], and locally inflamed DRG [97,98]. In addition, CXCL9 expression is also elevated in the joints of patients with rheumatoid arthritis [99]. Importantly, intrathecally injected (±)-NBI 74330 diminished the spinal level of the *CXCL9* mRNA [22]. In summary, an intrathecally administered antagonist of CXCR2 (NVP CXCR2 20) was effective at reducing the levels of *CCL2, CCL6, CCL7,* and *CXCL4*; in contrast, an antagonist of CXCR3 ((±)-NBI 74330) diminished levels of the *CCL3, CCL6, CXCL4,* and *CXCL9* mRNAs, which indicates slightly different mechanisms of action. Our *in vitro* experiments were designed to clarify the potential direct effects of CXCR2 and CXCR3 signaling on glial cells, especially since microglia play an important role in the development of neuropathy [68,100,101,102]. Moreover, our previous *in vitro* results suggest that CCL2, CCL3, CCL4, CXCL4, CXCL9, and CXCL10 may originate from microglial cells [8,13,22]. Additionally, in the present study, we confirmed the increase in mRNA levels of these chemokines. Additionally, for the first time, we described the increased production of CCL6 in microglia. In contrast, the expression of CXCL4 and CCL6 was not detectable in astrocytes. The LPS-induced increase in the expression of CCL6-7, CXCL4, and CXCL9-10 was significantly suppressed by both antagonists in primary microglial cell cultures; however, (±)-NBI 74330 has a broader spectrum of activity, as it also reduced the levels of CCL2-4. We observed a significantly lower effect of CXCR2 and CXCR3 antagonists on the modulation of these factors in stimulated astrocytes; both antagonists reduced the level of CXCL9, while (±)-NBI 74330 also reduced the level of CCL4. The results obtained from primary glial cell cultures confirm findings from our earlier studies [22] on the modulation of microglial activity by (±)-NBI 74330 through the modulation of factors released by these cells.

These results have prompted us to continue our research since both (±)-NBI 74330 and NVP CXCR2 20 appear to be promising treatments for relieving pain. However, only the CXCR2 antagonist attenuated mechanical and thermal hypersensitivity in CCI-exposed mice after intraperitoneal injection. To our knowledge, the obtained results show beneficial analgesic effects of a peripherally injected CXCR2 antagonist for the first time; therefore, these findings may have important clinical implications. Thus, further in-depth studies are needed to understand the pharmacodynamic and pharmacokinetic profiles of NVP CXCR2 20 after intraperitoneal administration. In contrast, (±)-NBI 74330 did not induce analgesia when administered intraperitoneally.

## 4. Materials and Methods

### 4.1. Animals

Male Wistar rats (275–300 g) and Albino-Swiss mice (20–25 g) were used to perform the study. Rodents were provided by Charles River Laboratories International, Inc. (Sulzfeld, Germany) and housed in cages with sawdust bedding on a standard 12 h/12 h light/dark cycle (lights on at 06.00 a.m.), with food and water available ad libitum. Each species was located in a separate room. Experiments were carried out according to the recommendations and standards of the International Association for the Study of Pain (IASP) [103] and the National Institutes of Health (NIH) Guide for the Care and Use of Laboratory Animals and were approved by the Ethical Committee of the Maj Institute of Pharmacology of the Polish Academy of Sciences (1277/2015, 262/2017, 257/2019, and 235/2020). Care was taken to minimize animal suffering and reduce the number of animals used (3R policy). Animal studies are reported in compliance with the ARRIVE guidelines [104].

### 4.2. Catheter Implantation

Rats were implanted with catheters for intrathecal (*i.t.*) injections using the method reported by Yaksh and Rudy [105]. Prior to surgery, animals were anesthetized intraperitoneally (*i.p.*) with sodium pentobarbital (60 mg/kg). Thirteen cm-long polyethylene catheters (PE 10, Intramedic; Clay Adams, Parsippany, NJ, USA) with a dead space of 10 μL were immersed in 70% (*v*/*v*) ethanol and fully washed with water for injection before surgery. During insertion, 7.8 cm of each catheter was slowly introduced through the atlanto-occipital membrane to the subarachnoid space at the rostral level of the spinal cord lumbar enlargement (L4–L5). After the procedure, water (10 μL) was injected, and the catheters were tightened. The rats were allowed to recover for 1 week before further experimental procedures. No catheters were implanted in mice.

### 4.3. Chronic Constriction Injury (CCI)

Chronic constriction injury of the sciatic nerve was performed in rats under sodium pentobarbital anesthesia and in mice under isoflurane anesthesia using the procedure described by Bennett and Xie [106] and our previously published papers [21,107]. An incision was made below the hip bone, and the *biceps femoris* and *gluteus superficialis* were separated. The right sciatic nerve was exposed and loosely tied three times in mice and four times in rats using ligatures (4/0 silk) at 1-mm intervals until a brief twitch in the operated limb was observed.

### 4.4. Drug Administration

We used (±)-NBI 74330 (NBI, Tocris, Bristol, UK) and NVP CXCR2 20 (NVP, Tocris, Bristol, UK), which were dissolved in dimethyl sulfoxide (DMSO), to perform the study. The control group received DMSO (vehicle, V) on the same schedule. The substances were injected intrathecally (*i.t.*) or intraperitoneally (*i.p.*). Therefore, the study was divided into 3 sets of experiments. Single intrathecal CXCR2 and CXCR3 antagonist administration in rats: Single doses of V, (±)-NBI 74330 or NVP CXCR2 20 (10, 20, and 30 μg/5 μL, i.t.) were administered on the 7th day after CCI, and behavioral tests were performed 0.5, 1, 2, 4, 6, and 24 h after substance administration. Repeated intrathecal CXCR2 and CXCR3 antagonist administration in rats: Animals received V, (±)-NBI 74330, or NVP CXCR2 20 (10 μg/5 μL, i.t.) pre-emptively 16 and 1 h before the CCI procedure and then once a day for the next 7 days; on the last day of the experiment, the behavioral tests were performed 1 h after substance administration. Single intraperitoneal CXCR2 and CXCR3 antagonist administration in mice: Single doses of V, (±)-NBI 74330 or NVP CXCR2 20 (1, 5, and 10 mg/kg, *i.p*.) were administered on the 14th day after CCI, and behavioral tests were performed 1, 3, and 5 h after substance administration.

### 4.5. Behavioral Tests

#### 4.5.1. Mechanical Hypersensitivity Measurement (von Frey Test)

The von Frey apparatus (Dynamic Plantar Anesthesiometer, Cat. No. 37400, Ugo Basile, Italy) was used to measure mechanical hypersensitivity in rats, with filament strengths up to 26 g, as previously described [8,12,108]. Calibrated nylon monofilaments with 0.6–6 g of strength (Stoelting, Wood Dale, IL, USA) were used to measure hypersensitivity in mice, as described in our previous studies [18,59]. The rodents were placed in plastic cages with a wire-net floor 5 min before the experiment and were able to move freely on the surface. The touch stimulator of the instrument or filaments were moved under the operated limb, and the reaction of the animal to the stimulus was measured automatically (for rats) or until the animal lifted the injured paw (for mice). In naive rats, the reaction of both hind paws was measured.

#### 4.5.2. Thermal Hypersensitivity Measurement (Cold Plate Test)

Thermal hypersensitivity was measured using a cold plate apparatus (Cold/Hot Plate Analgesia Meter No. 05044 Columbus Instruments, USA and Cold Plate Analgesia meter Ugo Basile, Gemonio, Italy, for rats and mice, respectively) as previously described [21,109]. The animals were placed on the surface (5 °C for rats; 2 °C for mice) and held there until they lifted the operated hind paw. The cutoff latency for both species was 30 s. In naive rats, both hind paws were observed simultaneously.

### 4.6. Primary Microglial and Astroglial Cell Cultures

*In vitro* studies were performed using primary microglial and astroglial cell cultures prepared from the cerebral cortex of newborn Wistar rats as described in our previously published papers [12,110]. Cells were seeded at a density of 3 × 10^5^ cells/cm^2^ in culture medium consisting of high-glucose GlutaMAX™ DMEM supplemented with 10% heat-inactivated fetal bovine serum, 0.1 mg/mL streptomycin, and 100 U/mL penicillin (Gibco, New York, NY, USA) on poly-L-lysine-coated 75-cm^2^ culture flasks. The cells were grown at 37 °C in an incubator with a humidified atmosphere of 5% CO_2_ in air. Four and 12 days later, the culture medium was changed to fresh medium. Next, the loosely attached microglial cells were harvested by gentle shaking (70 rpm for 1 h and 90 rpm for 15 min) and centrifugation (800 rpm for 10 min). Cell viability was determined using trypan blue staining (Bio-Rad, Warsaw, Poland). Afterwards, fresh medium was added to the same culture bottles to obtain astroglial cells. Three days later, astrocytes were trypsinized using a 0.05% trypsin–EDTA solution (Sigma-Aldrich, St. Louis, MO, USA). Both microglial and astroglial cells were seeded on 24-well plates (for mRNA analysis) and 6-well plates (for protein analysis) at a density of 1.2 × 10^6^ cells/well. Next, microglial and astroglial cells were treated with (±)-NBI 74330 [NBI; 100 nM] or NVP CXCR2 20 [NVP; 100 nM] 30 min before LPS (lipopolysaccharide from *Escherichia coli* 0111:B4; Sigma-Aldrich, St. Louis, MO, USA) administration (100 ng/mL) or harvested without LPS stimulation. IBA-1 and GFAP markers were used to visualize cell purity.

### 4.7. Analysis of Gene Expression (RT-qPCR)

Samples of ipsilateral lumbar segments of the spinal cord (L4–L6) and the DRG (L4–L6 pooled into one sample) were collected after the decapitation of naive and CCI-exposed rats on the 7th day after sciatic nerve injury, 4 h after the last V, (±)-NBI 74330, or NVP CXCR2 20 injection. Moreover, microglial and astroglial cell lysates were collected 24 h after LPS treatment. According to Chomczynski and Sacchi [111], total RNA extraction was performed using TRIzol reagent (Invitrogen, Carlsbad, CA, USA). The concentration and quality of RNA were measured using a DeNovix DS-11 Spectrophotometer (DeNovix Inc., Wilmington, NC, USA). An Omniscript RT Kit (Qiagen Inc., Hilden, Germany), oligo (dT16) primer (Qiagen Inc., Hilden, Germany), and RNAse inhibitor (rRNasin, Promega, Mannheim, Germany) were used to perform reverse transcription of 1 μg of total RNA from the tissue and 0.5 μg of total RNA from the cells at 37 °C. The obtained cDNA templates were diluted 1:10 using RNase-/DNase-free H_2_O. RT-qPCR was conducted with approximately 50 ng of cDNA templates from each sample using Assay-On-Demand TaqMan probes (Applied Biosystems, Foster City, CA, USA) and an iCycler device (Bio-Rad, Hercules, Warsaw, Poland). The following TaqMan primers were used: *IL-1β*, Rn00566700_m1; *IL-18*, Rn01422083_m1; *IL-6*, Rn01410330_m1; *CCL2*, Rn00580555_m1; *CCL3*, Rn00564660_m1; *CCL4*, Rn00671924_m1; *CCL6*, Rn01456400_m1; *CCL7*, Rn01467286_m1; *CXCL4*, Rn01768297_g1; *CXCL9*, Rn00595504_m1; and *CXCL10*, Rn00594648_m1. The cycle threshold values were automatically calculated with CFX Manager v.2.1 software using the default parameters. The RNA content was calculated using the formula 2−∆∆CT. *HPRT,* Rn01527838_g1, was used as a control and an adequate housekeeping gene, as previously described [21].

### 4.8. Analysis of Protein Levels (Western Blot)

Samples from the ipsilateral spinal cord (L4–L6) and DRG (L4–L6 polled into one sample) were collected 6 h after the last vehicle, (±)-NBI 74330 or NVP CXCR2 20 injection on the 7th day after CCI. Additionally, microglial and astroglial cell lysates were collected 1 h or 24 h after the LPS treatment. Next, tissue and cell lysates were placed in RIPA buffer supplemented with a protease inhibitor cocktail (Sigma-Aldrich, St. Louis, MO, USA), homogenized, and cleared via centrifugation (30 min, 14,000 rpm, 4 °C). The total protein concentration was measured using the bicinchoninic acid (BCA) method. The obtained samples (20 µg of protein) were mixed with loading buffer (4× Laemmli Buffer, Bio-Rad, Warsaw, Poland) and heated for 5 min at 96 °C. Electrophoresis was performed using 4–15% Criterion™ TGX™ precast polyacrylamide gels (Bio-Rad, Warsaw, Poland). Next, the proteins were transferred (semidry transfer 30 min, 25 V) to Immune-Blot PVDF membranes (Bio-Rad, Warsaw, Poland) and then blocked for 1 h at RT with 5% nonfat, dry milk (Bio-Rad, Warsaw, Poland) in Tris-buffered saline containing 0.1% Tween-20 (TBST). Afterwards, the membranes were washed with TBST buffer and incubated overnight at 4 °C with the following primary antibodies: rabbit: anti-IL-1β (1:500, Abcam, Cambridge, UK), anti-IL-18 (1:1000, Abcam, Cambridge, UK), anti-IL-6 (1:500, Invitrogen, Carlsbad, CA, USA), anti-IL-18BP (1:500, Novus, Abingdon, UK), anti-IL-1RA (1:2000, Abcam, Cambridge, UK), anti-IL-10 (1:1000, Abcam, Cambridge, UK), anti-p38 MAPK (1:1000, Cell Signaling, Danvers, MA, USA), anti-p-p38 MAPK (1:1000, Cell Signaling, Danvers, MA, USA), anti-ERK1/2 (1:1000, Cell Signaling, Danvers, MA, USA), anti-p-ERK1/2 (1:1000 Cell Signaling, Danvers, MA, USA), and mouse anti-GAPDH (1:5000, Millipore, Darmstadt, Germany). The next day, the membranes were washed with TBST buffer and incubated for 1 h at RT with HRP-conjugated anti-rabbit or anti-mouse secondary antibodies (1:5000, Vector Laboratories, Burlingame, CA, USA). SignalBoost™ Immunoreaction Enhancer Kit (Merck Millipore Darmstadt, Germany) solution was used to dilute primary and secondary antibodies. Proteins were detected using Clarity™ Western ECL Substrate (Bio-Rad, Warsaw, Poland) and visualized using the Fujifilm LAS-4000 FluorImager system. Fujifilm MULTI GAUGE software was used to estimate the levels of immunoreactive bands.

### 4.9. Analysis of Protein Levels (MILLIPLEX^®^ Multiplex Assays Using Luminex^®^)

Samples from the dorsal lumbar (L4–L6) spinal cord were collected 7 days after CCI and prepared for protein analysis in the same manner as described in the Western blot section. Concentrations of IL-1β, IL-18, and IL-6 were measured using a MILLIPLEX^®^ MAP Rat Cytokine/Chemokine Magnetic Bead Panel Immunology Multiplex Assay (Merck Millipore, Burlington, MA, USA) according to the manufacturer’s instructions and recommendations.

### 4.10. Statistical Analysis

#### 4.10.1. Behavioral Analysis

For studies of dose- and time-dependent pain-like behaviors, the data are presented as the percentage of maximal possible effect (%MPE) = (postdrug response − basal response)/(cutoff value − basal response) × 100%) ± SEM (Figure 1A–D and Figure 8A–D). Differences in animals administered different doses and at different time points were analyzed using two-way analysis of variance (ANOVA), followed by Bonferroni’s post hoc test for multiple comparisons. The overall effect and comparison of the effects on behavioral performance and the area under the curve (AUC) of antinociceptive effects (Figure 1E,F and Figure 8E,F) were calculated using trapezoidal and Simpson’s rules, as described previously [107]. For studies with repeated drug administration, the data are presented as the means ± SEM in grams or seconds (Figure 2A,B). The intergroup differences were analyzed using one-way analysis of variance (ANOVA) followed by Bonferroni’s post hoc test.

#### 4.10.2. Biochemical Analysis

The results of RT-qPCR, Western blot, and multiplex assays using Luminex analysis are presented as fold changes relative to the control (naive) group ± SEM (Figure 3, Figure 4, Figure 5, Figure 6 and Figure 7). The data were analyzed using one-way analysis of variance (ANOVA) followed by Bonferroni’s post hoc test. The results of RT-qPCR and Western blot analyses of samples from *in vitro* studies are presented as fold changes relative to the control group (V, LPS- or LPS+) ± SEM. The data were analyzed using one-way analysis of variance (ANOVA) followed by Bonferroni’s post hoc test. All data were analyzed using GraphPad Prism 7 software.

## 5. Conclusions

Our research summarizes the neuroimmune crosstalk in which the CXCR2 and CXCR3 receptors are involved as an important signaling mechanism during nociceptive transmission in neuropathic pain. Nevertheless, in the behavioral tests, NVP CXCR2 20 (a CXCR2 antagonist) and (±)-NBI 74330 (a CXCR3 antagonist) showed diverse potential for promoting pain relief. In addition to common components, biochemical analyses allowed us to identify their different mechanisms of action in the modulation of interleukins, chemokines, and intracellular signaling pathways. The CXCR3 antagonist (±)-NBI 74330 has a broader spectrum of action after intrathecal administration, while NVP CXCR2 20 also alleviated neuropathic pain after peripheral administration. Undoubtedly, these findings require more detailed research; however, we suggest that the blockade of CXCR2 may represent a new strategy for neuropathic pain therapy in the future.

## Figures and Tables

**Figure 1 ijms-22-11074-f001:**
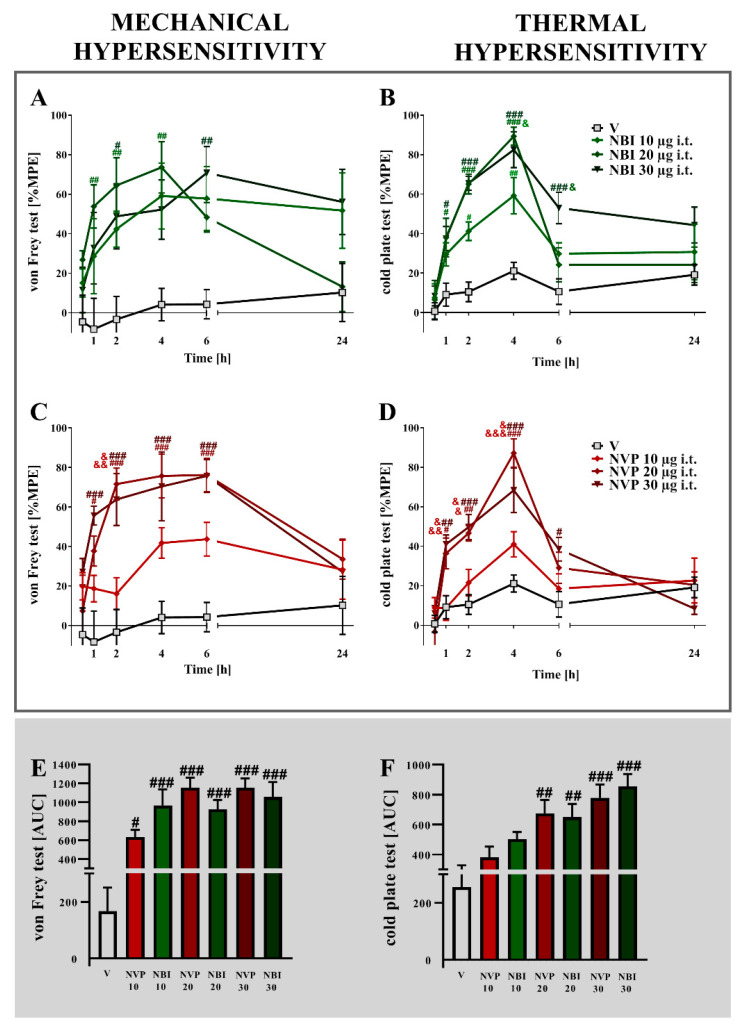
Comparison of the analgesic effects of intrathecal (*i.t*.) administration of different doses (10, 20, or 30 µg/5 μL) of (±)-NBI 74330 (**A**,**B**) and NVP CXCR2 20 (**C**,**D**) on the 7th day after chronic constriction injury in rats. The effects of single *i.t.* NBI and NVP injections on mechanical (**A**,**C**) and thermal (**B**,**D**) hypersensitivity were measured. The results are presented as a percentage of the maximal possible effect (% MPE = (post-drug response − basal response)/(cutoff value − basal response) × 100%) of drug action (**A**–**D**), and the area under the curve (AUC) for each test was calculated (**E**,**F**). Data are presented as the means ± SEM (4–6 rats per group). The results were evaluated using one-way ANOVA followed by Bonferroni’s test for comparisons of selected pairs measured separately at each time point. Additionally, the results were evaluated using two-way ANOVA to determine the time × drug interaction. # *p* < 0.05, ## *p* < 0.01, and ### *p* < 0.001 for the comparison of vehicle-treated animals with all groups at the indicated time points. & *p* < 0.05, && *p* < 0.01, and &&& *p* < 0.001 for the comparison of different NBI or NVP doses, and the same doses of NVP with NBI. Abbreviations: vehicle, V; (±)-NBI 74330, NBI; NVP CXCR2 20, NVP.

**Figure 2 ijms-22-11074-f002:**
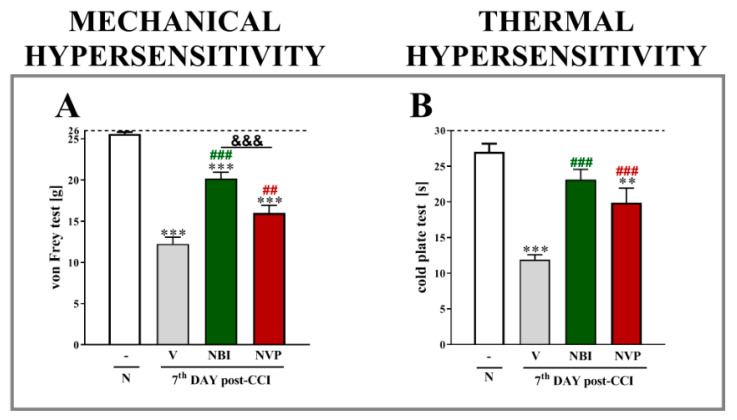
Comparison of the analgesic effects of repeated intrathecal (*i.t*.) administration of (±)-NBI 74330 and NVP CXCR2 20 (10 µg/5 μL, each; (**A**,**B**)) on the 7th day after chronic constriction injury in rats. The effects of repeated *i.t.* NBI and NVP injection on mechanical (**A**) and thermal (**B**) hypersensitivity were measured at 120 and 125 min after the last drug administration, respectively. The data are presented as the means ± SEM (10 rats per group), and the horizontal dotted line shows the cutoff value. The results were evaluated using one-way ANOVA followed by Bonferroni’s test. ** *p* < 0.01 and *** *p* < 0.001 for the comparison of naive animals with all groups. ## *p* < 0.01 and ### *p* < 0.001 for the comparison of vehicle-treated animals with all groups. &&& *p* < 0.001 for the comparison of animals treated with NBI or NVP. Abbreviations: naive, N; vehicle, V; (±)-NBI 74330, NBI; NVP CXCR2 20, NVP.

**Figure 3 ijms-22-11074-f003:**
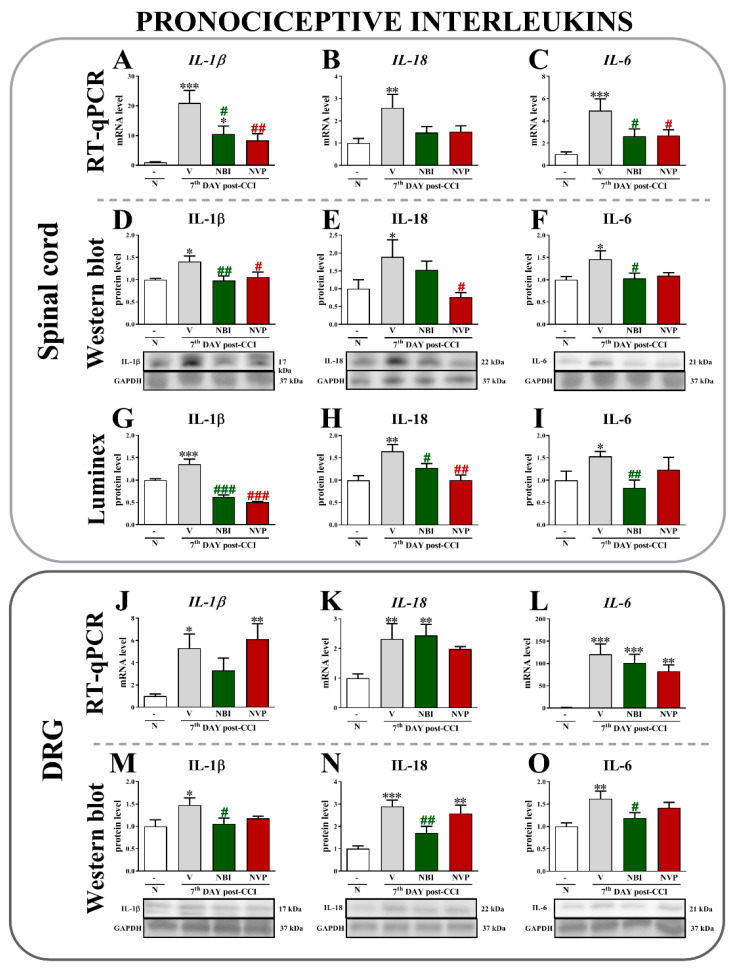
Comparison of the effects of repeated intrathecal (*i.t*.) administration of (±)-NBI 74330 and NVP CXCR2 20 (10 µg/5 μL; each) on the mRNA (**A**–**C**,**J**–**L**) and protein (**D**–**I**,**M**–**O**) levels of select interleukins (IL-1β, IL-18, and IL-6) in the spinal cord (**A**–**I**) and DRG (**J**–**O**) on the 7th day after chronic constriction injury in rats. The RT-qPCR, Western blot, and Luminex data are presented as the means ± SEM (4–10 rats per group for RT-qPCR and 4–7 for Western blot or Luminex assays). The results were evaluated using one-way ANOVA followed by Bonferroni’s test. * *p* < 0.05, ** *p* < 0.01, and *** *p* < 0.001 for the comparison of naive animals with all groups. # *p* < 0.05, ## *p* < 0.01, and ### *p* < 0.001 for the comparison of vehicle-treated animals with all groups. Abbreviations: naive, N; vehicle, V; (±)-NBI 74330, NBI; NVP CXCR2 20, NVP.

**Figure 4 ijms-22-11074-f004:**
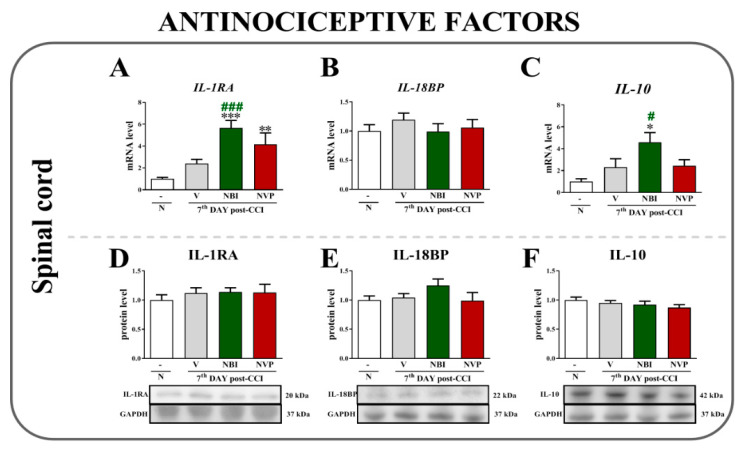
Comparison of the effects of repeated intrathecal (*i.t*.) administration of (±)-NBI 74330 and NVP CXCR2 20 (10 µg/5 μL; each) on the mRNA (**A**–**C**) and protein (**D**–**F**) levels of select antinociceptive factors (IL-1RA, IL-18BP, and IL-10) in the spinal cord (**A**–**F**) on the 7th day after chronic constriction injury in rats. The RT-qPCR and Western blot data are presented as the means ± SEM (4–9 rats per group for RT-qPCR and 5–7 for Western blot). The results were evaluated using one-way ANOVA followed by Bonferroni’s test. * *p* < 0.05, ** *p* < 0.01, and *** *p* < 0.001 for the comparison of naive animals with all groups. # *p* < 0.05 and ### *p* < 0.001 for the comparison of vehicle-treated animals with all groups. Abbreviations: naive, N; vehicle, V; (±)-NBI 74330, NBI; NVP CXCR2 20, NVP.

**Figure 5 ijms-22-11074-f005:**
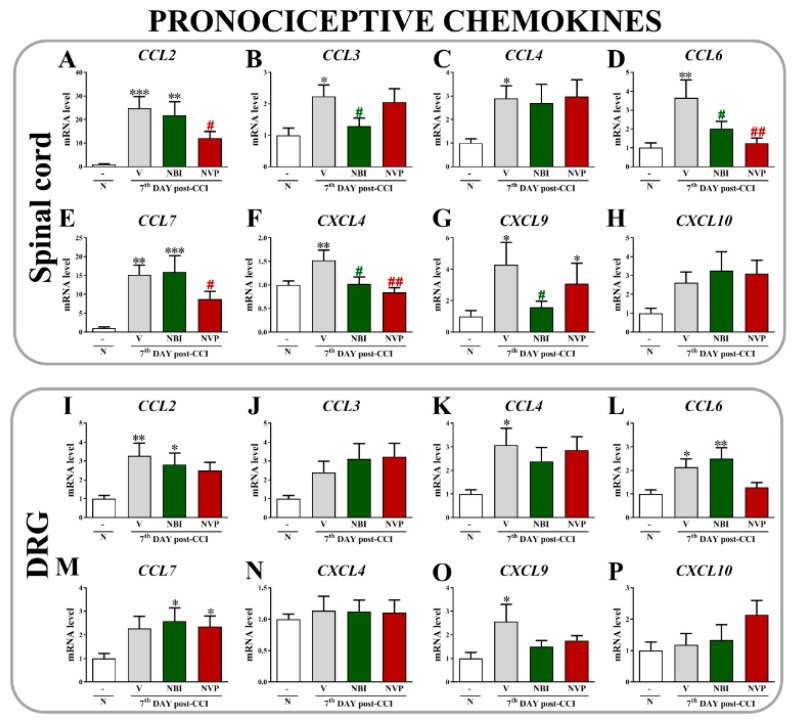
Comparison of the effects of repeated intrathecal (*i.t*.) administration of (±)-NBI 74330 and NVP CXCR2 20 (10 µg/5 μL; each) on the mRNA (**A**–**P**) levels of select chemokines (CCL2, CCL3, CCL4, CCL6, CCL7, CXCL4, CXCL9, and CXCL10) in the spinal cord (**A**–**H**) and DRG (**I**–**P**) on the 7th day after chronic constriction injury in rats. The RT-qPCR data are presented as the means ± SEM (4–9 rats per group). The results were evaluated using one-way ANOVA followed by Bonferroni’s test. * *p* < 0.05, ** *p* < 0.01, and *** *p* < 0.001 for the comparison of naive animals with all groups. # *p* < 0.05 and ## *p* < 0.01 for the comparison of vehicle-treated animals with all groups. Abbreviations: naive, N; vehicle, V; (±)-NBI 74330, NBI; NVP CXCR2 20, NVP.

**Figure 6 ijms-22-11074-f006:**
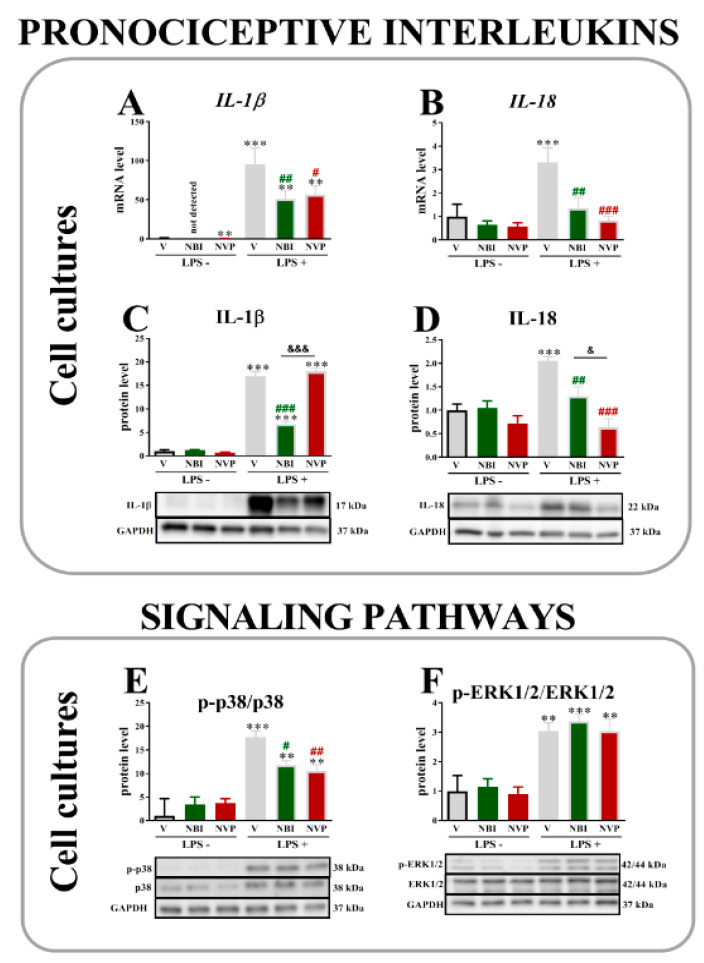
Comparison of the effects of (±)-NBI 74330 and NVP CXCR2 20 (100 nM; each) on the mRNA (**A**,**B**) and protein (**C**–**F**) levels of select interleukins (IL-1β and IL-18) and signaling pathways (p38 and ERK1/2) in microglial (**A**–**F**) cell cultures. The RT-qPCR and Western blot data are presented as the means ± SEM (3–4 independent experiments). The results were evaluated using one-way ANOVA followed by Bonferroni’s test. ** *p* < 0.01 and *** *p* < 0.001 for the comparison of the control group (vehicle-treated nonstimulated cells) with all groups. # *p* < 0.05, ## *p* < 0.01, and ### *p* < 0.001 for the comparison of vehicle-treated LPS-stimulated cells with all groups. & *p* < 0.05 and &&& *p* < 0.001 for the comparison of NBI with NVP groups. Abbreviations: vehicle, V; (±)-NBI 74330, NBI; NVP CXCR2 20, NVP; lipopolysaccharide, LPS; LPS+, LPS-stimulated cells; LPS-, LPS-nonstimulated cells.

**Figure 7 ijms-22-11074-f007:**
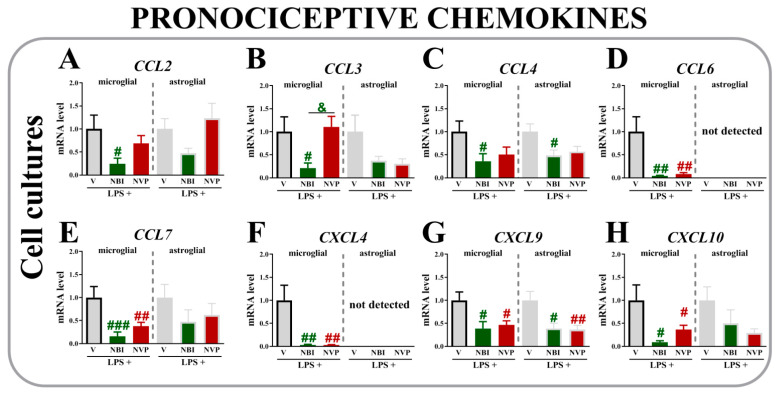
Comparison of the effects of (±)-NBI 74330 and NVP CXCR2 20 (100 nM; each; (**A**–**H**)) on the mRNA levels of select chemokines (CCL2, CCL3, CCL4, CCL6, CCL7, CXCL4, CXCL9, and CXCL10) in microglial (left panel) and astroglial (right panel) cell cultures. The RT-qPCR data are presented as the means ± SEM (3–4 independent experiments). The results were evaluated using one-way ANOVA followed by Bonferroni’s test. # *p* < 0.05; ## *p* < 0.01 and ### *p* < 0.001 for the comparison of vehicle-treated LPS-stimulated cells with all groups. & *p* < 0.05 for the comparison of NBI with NVP groups. Abbreviations: vehicle, V; (±)-NBI 74330, NBI; NVP CXCR2 20, NVP; lipopolysaccharide, LPS. LPS+, LPS-stimulated cells; LPS-, LPS-nonstimulated cells.

**Figure 8 ijms-22-11074-f008:**
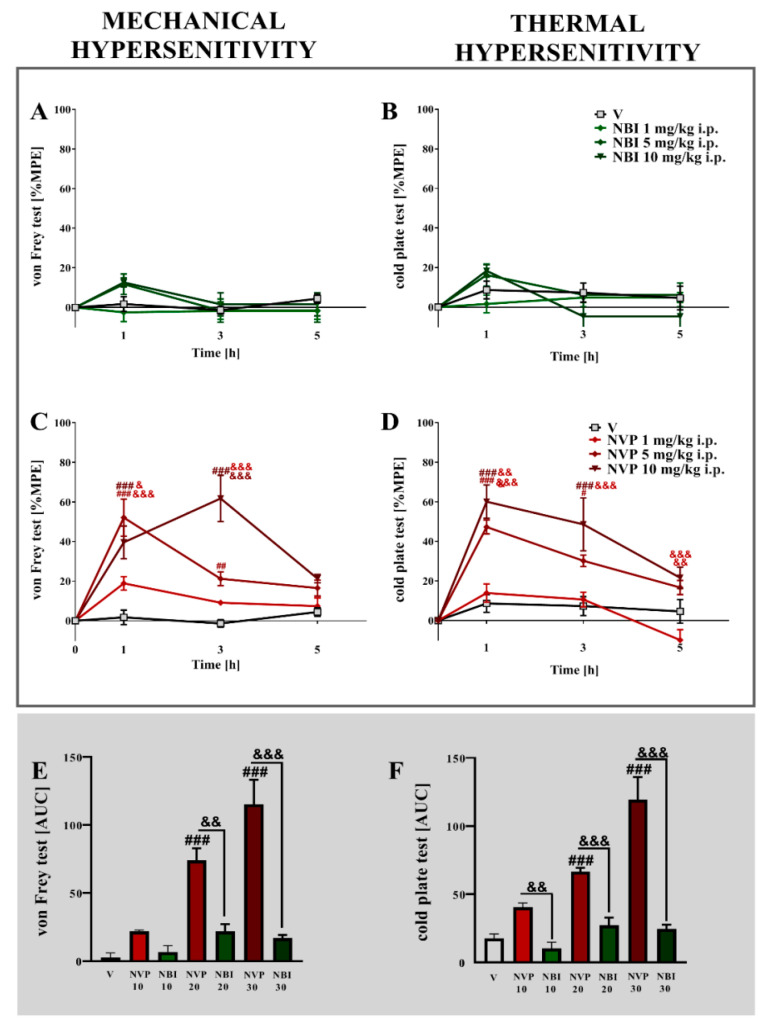
Comparison of the analgesic effects of intraperitoneal (*i.p*.) administration of different doses (1, 5, or 10 mg/kg) of (±)-NBI 74330 (**A**,**B**) and NVP CXCR2 20 (**C**,**D**) on the 7th day after chronic constriction injury in mice. The effects of single *i.p*. NBI and NVP injections on mechanical (**A**,**C**) and thermal (**B**,**D**) hypersensitivity were measured. The results are presented as a percentage of the maximal possible effect (% MPE = (post-drug response − basal response)/(cutoff value − basal response) × 100%) of drug action (**A**–**D**), and the area under the curve (AUC) for each test was calculated (**E**,**F**). Data are presented as the means ± SEM (4–6 mice per group). The results were evaluated using one-way ANOVA followed by Bonferroni’s test for comparisons of selected pairs measured separately at each time point. Additionally, the results were evaluated using two-way ANOVA to determine the time × drug interaction. # *p* < 0.05 and ### *p* < 0.001 for the comparison of vehicle-treated animals with all groups at the indicated time points. & *p* < 0.05, && *p* < 0.01, and &&& *p* < 0.001 for the comparison of different doses of NVP, and the same doses of NVP with NBI. *Abbreviations:* vehicle, V; (±)-NBI 74330, NBI; NVP CXCR2 20, NVP.

## Data Availability

The data presented in this study are available on request from the corresponding authors.

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
