# Peer review of "Comparison of the Effects of Chemokine Receptors CXCR2 and CXCR3 Pharmacological Modulation in Neuropathic Pain Model—In Vivo and In Vitro Study"

_ijms, 2021, doi:10.3390/ijms222011074_

Round 1

Reviewer 1 Report

The Authors, Drs Piotrowska et al., submitted an article in which they provide evidence that single/repeated intrathecal administration of chemokines CXCR2-(NVP-CXCR2-20) and CXCR3-((±)-NBI-74330)) antagonists explicitly attenuated mechanical/thermal hypersensitivity in rats after chronic constriction injury of the sciatic nerve. Their research revealed important roles for CXCR2 and CXCR3 in nociceptive transmission, especially in neuropathic pain

The manuscript is scientifically interesting. Paragraphs are well organized, and the purpose of the article is clear. However, the title is too long and the English language needs revisions.

Author Response

Thank you for appreciating our work - we are very grateful for Your review.

According to your suggestion we have shortened the title of the manuscript

from

Comparison of the effects of pharmacological modulation of two CXC chemokine receptors, CXCR2 and CXCR3, on nociceptive processes in neuropathic pain model - in vivo and in vitro evidence”.

into

“Comparison of the effects of chemokine receptors CXCR2 and CXCR3 pharmacological modulation in neuropathic pain model - in vivo and in vitro study”

Moreover, we would like to inform that the English language was corrected by American Journal Experts (certificate no 3D77-CE59-5BF1-B49A-CC15). We have made every effort to remove linguistic errors.

Reviewer 2 Report

CXC chemokines have been involved in the mechanisms of neuropathic pain. This study aims to mitigate hypersensitivity to chronic constriction injury in rats and mice by intrathecal administration of antagonists of CXCR2 and CXCR3.

Authors found that single or repeated intrathecal administration of CXCR2 and CXCR3 antagonists reduced hypersensitivity in a rat model of chronic constriction injury. This correlated with significant reduction of expression of several pro-inflammatory cytokines in the spinal cord and dorsal root ganglia. Blocking CXCR2 and CXCR3 also reduced the expression of several sensory factors that are upregulated by chronic constriction injury. This may be caused by a reduced activity of the p38 MAPK pathway, but not via ERK1/2.

Data is interesting and relevant to neuropathic pain. However, there are some major concerns in this study:

-           Figure 5 and 7 are identical, and it was not possible to evaluate the in vitro part of the study.

-           All figures show data for qPCR and Western blot, however, Figure 5 lacks of Western blot analyses. Adding Western blots of some chemokines would maintain a consistency along the manuscript and could help to a better interpretation of data.

-           Investigating the role of CXCR2 and CXCR3 antagonists in rats and mice is laudable. However, data from mice is limited as only limits to behavior analyses and not to molecular analyses, as authors show for the rat model. Furthermore, are the antagonists concentration used in rats equivalent to those used in mice? Why the method to deliver CXCR2 and CXCR3 antagonists are different between rats and mice?

-           If authors wanted to stablish a connection between the beneficial role of CXCR2 and CXCR3 antagonists and microglia/astrocytes, authors should show evidence that the state of microglia/astrocytes (reactive?) differs between control and treated animals.

Author Response

Thank you for reviewing and appreciating our manuscript. Based on your valuable remarks we have made careful modifications to the original paper.  All changes are marked in yellow.

Data is interesting and relevant to neuropathic pain. However, there are some major concerns in this study:

-           Figure 5 and 7 are identical, and it was not possible to evaluate the in vitro part of the study.

Thank you very much for this important remark - We would like to apologize, unfortunately we have pasted the figure 5 twice by mistake.  We have re-attached the manuscript with the proper figure 7.

-           All figures show data for qPCR and Western blot, however, Figure 5 lacks of Western blot analyses. Adding Western blots of some chemokines would maintain a consistency along the manuscript and could help to a better interpretation of data.

We agree with the Referee, however since the chemokines are very small cytokines we were not able to make the Western blot analysis. Moreover, since the chemokines are very similar to each other is also difficult to find reliable antibodies for Western blot analysis.

-           Investigating the role of CXCR2 and CXCR3 antagonists in rats and mice is laudable. However, data from mice is limited as only limits to behavior analyses and not to molecular analyses, as authors show for the rat model. Furthermore, are the antagonists concentration used in rats equivalent to those used in mice? Why the method to deliver CXCR2 and CXCR3 antagonists are different between rats and mice?

- We would like to explain that we have begun our study with CXCR2 and CXCR3 antagonists administered repeatedly intrathecally in neuropathic rats, which allowed us to investigate how blocking these receptors affects nociceptive transmission and nociceptive factors directly at the spinal cord and DRG levels. Additionally, the repeated intrathecal injections could not be performed in mice, since we are not able to perform catherers implantations in mice. Due to the strong analgesic effects of both antagonists administered intrathecally in rats, in the second part of our study we decided to give the antagonists intraperitoneally on already developed neuropathic pain, because this route and scheme of drug administration better reflects the clinical situation. The antagonists concentrations could not be identical in mice and rats, because we are dealing with completely different routes of administration. Importantly, only the CXCR2 antagonist attenuated mechanical and thermal hypersensitivity in CCI-exposed mice after single intraperitoneal injection. We agree that it is interesting result, therefore in the near future we would like to continue the study – we are planning to make repeated CXCR2 antagonist intraperitoneal administration in neuropathic mice and according to your suggestion we will make the molecular analysis in several time points. However, this is a very large experiment that cannot be done in frame of 10 days.

-           If authors wanted to stablish a connection between the beneficial role of CXCR2 and CXCR3 antagonists and microglia/astrocytes, authors should show evidence that the state of microglia/astrocytes (reactive?) differs between control and treated animals.

We agree with the Referee, that such information in the discussion part is missing. It was happen probably because we have been studied the activation of microglial and astroglial cells in a mouse and rat model of neuropathic pain for years, so it was for us so obvious that we forgot to introduced such information into the text  – thank you very much for your comment. Moreover, in our previous published papers in 2019 - Frontiers in Immunology(doi:10.3389/fimmu.2019.02198 [21] ) and 2018 BBA-Molecular Basis of Disease (doi:10.1016/j.bbadis.2018.07.032 [22] ) we have reported co-localization of both receptors with microglia and astroglia markers in naive and CCI-exposed rats using immunofluorescence staining.

According to the Referee suggestion we have added into discussion part the following sentence: “Under neuropathic pain, we have previously shown that microglia and astroglia cells are strongly activated [8, 17, 57] and together with others we proved that both CXCR2 and CXCR3 are also present on these cells [21,22,34,44,52].”

Round 2

Reviewer 2 Report

Authors have replied the reviewer's comments satisfactorily.

Thank you.